# End of treatment and 12-month post-treatment outcomes in patients treated with all-oral regimens for rifampicin-resistant tuberculosis in Ukraine: a prospective cohort study

Vini Fardhdiani[1,2], Olena Trush[1], Nataliia Lytvynenko[3], Svitlana Pylypchuk[3], Yana Terleeva[4], Khachatur Malakyan[5], Praharshinie Rupasinghe[6,7], Yves Wally[2], Marve Duka[1], Vitaly Stephanovich Didyk[8], Olga Valentinovna Siomak[8], Oleksandr Blyzniuk[8], Jennifer Furin[9], Dmytro Donchuk[10], Chinmay Laxmeshwar[1,11], Petros Isaakidis[10,12]*

**1** Médecins Sans Frontières, Zhytomyr, Ukraine, **2** Médecins Sans Frontières, Brussels, Belgium, **3** National TB Institute, Kyiv, Ukraine, **4** Department of TB Programme Coordination, Public Health Center of the MOH, Kyiv, Ukraine, **5** Médecins Sans Frontières, Kyiv, Ukraine, **6** Mycobacteriology unit, Department of Biomedical Sciences, Institute of Tropical Medicine, Antwerp, Belgium, **7** Department of Biomedical Sciences, University of Antwerp, Antwerp, Belgium, **8** Zhytomyr Regional TB Dispensary, Zhytomyr, Ukraine, **9** Harvard Medical School, Cambridge, Massachusetts, United States of America, **10** Southern Africa Medical Unit, Médecins Sans Frontières, Cape Town, South Africa, **11** Department of Global Health, University of Washington, Seattle, United States of America, **12** Clinical and Molecular Epidemiology Unit, Department of Hygiene and Epidemiology, University of Ioannina School of Medicine, Ioannina, Greece

* petros.isaakidis@joburg.msf.org

## Abstract

The World Health Organization has called for operational research on all-oral shorter regimens for rifampin-resistant and multidrug-resistant forms of tuberculosis (RR/MDR-TB). We followed a cohort of patients in Zhytomyr, Ukraine for effectiveness, safety, tolerability and feasibility of bedaquiline & delamanid-based treatment regimens under programmatic conditions. This was a single-arm implementation study. All consenting persons with RR/MDR-TB were enrolled between 1 April 2019 and 31 May 2021 and followed up 12-months after treatment completion. We assessed quality of life and depression symptoms between start and end-of-treatment. We enrolled 300 patients. Overall, 212 (71%) patients were cured, 22 (7%) patients completed treatment, median time to culture conversion was 58 days (IQR:30–75), and 21% and 27% of patients had at least one serious or Grade 3/4 adverse event, respectively. The overall BREF-WHO/Quality of Life score improved between baseline and end-of-treatment, from average 52.64(std. dev:21.63) to 57.15(std. dev:21.43) while Patient Health Questionnaire-9(PHQ-9) score decreased from 6.67(std. dev:4.75) at baseline to 5.34(std. dev: 5.18) at end-of-treatment. Twelve months post-treatment 174/234(74%) were alive and recurrence-free, 17(7%) patients died, one (<1%) had recurrent TB, while 42 (18%) were lost from the post-treatment follow-up. All-oral short-term regimens showed high success under programmatic conditions in

**Data availability statement:** Data will be available on https://nexo.msf.org/. In case of technical issues requests should be made to data.sharing@msf.org. For more information please see: 1) MSF's Data Sharing Policy: https://www.msf.org/sites/msf.org/files/msf_data_sharing_policycontact_infoannexes_final.pdf 2) MSF's Data Sharing Policy PLOS Medicine article: https://journals.plos.org/plosmedicine/article?id=10.1371/journal.pmed.1001562.

**Funding:** Médecins Sans Frontières (MSF) provided support for this study in the form of salaries for VF, OT, SP, KM, YW, MD, VS, DD, CL, and PI. The specific roles of these authors are articulated in the 'author contributions' section. Research conducted by MSF employees is done so within the scope of their job specification. No other forms of support were provided specifically for the research. MSF was involved in study design, data collection, analysis, decision to publish and preparation of the manuscript.

**Competing interests:** The authors have read the journal's policy and have the following competing interests to declare: Médecins Sans Frontières (MSF) provided support in the form of salaries for VF, OT, SP, KM, YW, MD, VS, DD, CL, and PI. This does not alter our adherence to PLOS Global Public Health policies on sharing data and materials. There are no patents, products in development, or marketed products associated with this research.

Ukraine, despite extreme implementation challenges during the COVID-pandemic and the Russia-Ukraine war. Moreover, this was a cohort of patients with high levels of co-morbidities and substance use. A multidisciplinary, psychosocial support model might have contributed to satisfactory treatment outcomes, improved quality of life and decreased symptoms of depression among people living with RR/MDR-TB.

## Summary

We evaluated all-oral regimens for multidrug-resistant tuberculosis in Zhytomyr, Ukraine, achieving 78% success, despite high rates of co-infections and substance use. Seventy-four per-cent of patients remained relapse-free 12-months post-treatment, demonstrating feasibility under programmatic conditions during the COVID-19 pandemic and war.

## Introduction

Rifampicin-resistant/multidrug-resistant forms of TB (RR/MDR-TB) are a major global health concern, with nearly half a million people affected in 2022 [1]. As a rule, RR/MDR-TB regimens are lengthy and involve multiple toxic second-line agents, resulting in a success rate of just over 60% among those receiving them [2]. Although the past decade has seen remarkable promise in the treatment of RR/MDR-TB with effective new and re-purposed drugs being used successfully, global experience with regimens containing these drugs remains limited [3].

In 2019, while awaiting the completion of several randomized controlled trials aimed at optimizing RR/MDR-TB treatment, the World Health Organization (WHO) called for structured operational research to be carried out by TB programs on the use of all-oral, shorter regimens [4]. The recommendation aimed to promote implementation science, improving access to newer agents—including bedaquiline, delamanid, and linezolid—while enabling the structured collection of data on the performance of all-oral regimens in real-world conditions, particularly in settings and populations often excluded from formal clinical trials [5] Such operational research could also afford the opportunity to collect data on other aspects of RR/MDR-TB care—including person-centered approaches—that might be relevant during programmatic scale-up of newer regimens [6].

From 2019 to 2022, MSF, along with the National TB Institute and the Ministry of Health in Ukraine, conducted operational research to demonstrate a person-centered model of ambulatory care along with the use of all-oral short-course regimens for RR-TB in Zhytomyr oblast. This study took place during an extremely challenging period marked by national health reform, followed by the COVID-19 pandemic, and the Russia-Ukraine war. Here, we present the treatment outcomes, including 12-month recurrence-free survival, safety, tolerability, and feasibility of using all-oral short-course regimen as part of a comprehensive person-centered model of care.

## Methods

### Study design

We conducted an uncontrolled, single-arm clinical and feasibility study in the Zhytomyr Oblast of Ukraine. Enrollment of RR/MDR-TB patients commenced on 1 April 2019 and concluded on 31 May 2021. The treatment duration ranged from 9 to 12 months. The participants were monitored until 12 months after the successful completion of the treatment.

### Study setting

The study was conducted in the Zhytomyr oblast of Ukraine. In 2017, the city of Zhytomyr had a tuberculosis prevalence rate of 106.2 cases per 100,000 people, which was higher than the national average of 76.6 cases per 100,000 people. The oblast is served by a regional TB referral hospital located in Zhytomyr. This hospital also coordinated the ambulatory care for patients at outpatient facilities which included health posts, TB-units, ambulatory care facilities, family doctors, and polyclinics, and post-treatment follow-up. The health system for TB care in the oblast is presented elsewhere[7].

### RR-TB/MDR-TB treatment and monitoring protocol

Baseline assessments were conducted for all study participants at enrollment. These included bacteriological assessment (Xpert MTB/RIF® assay for all patients, if found RR by Xpert MTB/RIF® assay, Genotype MTBDRplus® MTBDRsl, culture and phenotypic DST), clinical assessment, laboratory assessment (for HIV, Hepatitis B, Hepatitis C, liver function test, renal function test, etc.), chest radiograph, and an electrocardiogram (ECG).

As per the national guidelines during the study period, all patients were hospitalized for the entire duration of their TB treatment. However, patients enrolled in this study were discharged to ambulatory care after achieving a smear-negative result. This approach contrasted with the national guidelines, which mandated prolonged hospitalizations. To ensure quality care during the ambulatory phase, the study team conducted an assessment of the outpatient facilities throughout the oblast and in close proximity with the patients' residences and provided support to strengthen the care available at these sites [7].

All patients received a treatment regimen that lasted for a period ranging from 9 to 12 months. All clinical decisions were made by the oblast level RR-TB consilium based out of the regional TB hospital in Zhytomyr. Individuals lacking any evidence of fluoroquinolone resistance were administered bedaquiline (BDQ), linezolid (LZD), levofloxacin (LFX), clofazimine (CFZ), and cycloserine (CS). Individuals with evidence of fluoroquinolone resistance were administered bedaquiline (BDQ), linezolid (LZD), delamanid (DLM), clofazimine (CFZ), and cycloserine (CS).

The first follow-up assessment occurred two weeks following treatment initiation. This encompassed the evaluation for adverse events and included an ECG for all. Subsequently, monthly follow-ups were conducted until the completion of treatment. The following procedures were conducted: sputum analysis for smear and culture, evaluation of adverse events, laboratory testing, ECG, assessment of adherence, provision of adherence support, and counselling. If a patient had a positive culture at or after three months of treatment, both baseline and follow-up isolates were subjected to phenotypic DST for new and repurposed drugs, including MIC testing if resistance to any new or repurposed drugs was detected.

The BREF WHO Quality of Life BREF (WHOQoL-BREF) and Patient Health Questionnaire (PHQ) 9 questionnaires were administered at baseline and then at months 3, 6, 9, and 12. HIV viral load testing was conducted at months 6 and 12, and CD4 count was performed at month 12 for those who tested positive for HIV.

Post-treatment follow-up assessments were conducted until 12 months after the completion of treatment. A nurse contacted the patients via home visits or telephone calls, while during the COVID-19 pandemic and post-war, this communication was mainly telephonic. After three failed attempts to contact a patient within one month, we considered them as post-treatment lost to follow-up.

An important element of this study was providing extensive and multidisciplinary patient support beyond the national guidelines. This encompassed various activities, such as treatment literacy, intensive adherence support and monitoring, psychosocial support during both hospitalization and ambulatory care, and community involvement (Box 1). All patients with alcohol use disorder were provided care by a psychiatrist and psychologist to enable them to continue their treatment. The alcohol use disorder (AUD) care package and its results are presented elsewhere [8]. All study procedures were conducted as routine care for RR/MDR-TB treatment in Zhytomyr by the Ministry of Health (MOH) staff with support from MSF.

**Box 1. Patient and community support for RR-TB patients in Zhytomyr Oblast, Ukraine, April 2019 - May 2021.**

| Activity | Resources | Expected Outcome |
| --- | --- | --- |
| **Patient support** | | |
| Treatment literacy after diagnosis | Dedicated staff (doctor or nurse) conducted a 20-minute session with the patient on the following topics: What is TB? How did I get this? How do I prevent others from getting it? Three subsequent counseling sessions for patients after initiation of treatment (hospitalization, 1st, 2nd, 3rd day) by nurse and social worker. One session with the family members conducted by a nurse and social worker before discharge. | Patients understand that anyone can be infected, and that TB is curable. Family members go for TB screening themselves. Patients learn about the TB disease and its transmission, medicines used for treatment, possible side effects, monitoring schedule, importance of adherence, support available during treatment. Sessions help to reduce the level of self-stigmatization. Family members understand the patient's predicament and help reduce/eliminate stigma; support the patient morally and emotionally. |
| Treatment monitoring and adherence support | Nurse (MoH[1]) with DOT[2], alternative involvement of ambulatory care facilities, NGO[3] (Let Your Heart Beat) along with a trusted family member of patient as the treatment buddy. VOT[4] and SAT[5] provided to patients who needed them. Regular follow-up and medical check-ups | Patients receive medication on time and can adhere to the treatment regimen |
| Psychosocial support during hospitalization and ambulatory care | Social worker conducted patients' needs assessment; actions based on patients' needs. Provision of psychological support, if needed. Budget for social needs: support to apply for public aid like unemployment benefits, transportation reimbursements to attend follow-ups, food and hygiene articles support, firewood support during winter. Renewal of patients' needs assessment if necessary (constantly during treatment) | Patients were psychologically supported throughout the treatment process. Basic social needs are satisfied. Patients have all necessary documents to apply for government funding and to be independent from other external support. |
| Assess AUD[6], SUD[7], depression | Psychologist/nurse assessed/screened for mental health needs and AUD[6] Provided support for linkage with public psychiatric services | Comprehensive management of patient's medical condition improves quality of life and leads to successful TB treatment outcome |
| **Community Support** | | |
| Involving communities where patients originate from | Trained health workers and social workers conduct awareness sessions with neighbors and friends. | Reduction in stigmatizing behavior, additional support from the community to the patient |

[1]MoH: Ministry of Health; [2]DOT: Direct Observed Therapy; [3]NGO: Non-Governmental Organisation; [4]VOT: Video Observed Therapy; [5]SAT: Self-Administered Therapy; [6]AUD: Alcohol Use Disorder; [7]SUD: Substance Use Disorder.

**Adaptations to implementation during COVID-19.** The COVID-19 pandemic significantly restricted the interactions between mobile teams and patients undergoing ambulatory treatment. Consequently, all psychological counseling and support sessions were conducted via phone calls. Additionally, nurses incorporated infection control education into their

sessions and distributed personal protective equipment (PPE). To minimize contact, only one mobile team member interacted directly with the patient while distributing social support packages. Despite these challenges, all services outlined in the treatment protocol were delivered timely. Upon discharge from the hospital, transportation for patients to their homes using taxis was provided. This measure aimed to prevent patients from exposure to potentially infected individuals in public transportation.

**Adaptations to implementation post-war period.** The onset of the war had a detrimental impact on the ambulatory treatment of TB patients. Public transportation became inaccessible for many, and military checkpoints on roads made it extremely difficult or even impossible for patients to reach medical facilities. Additionally, the closure of stores led to a shortage of essential products and hygiene items for patients. During the war, we continued to provide continuous support through a multidisciplinary team comprising a nurse, social worker, and psychologist. The contents of the monthly food packages provided to patients were increased with an aim to cover 70% of their nutritional needs. The MSF mobile teams also facilitated the transportation of sputum samples to the laboratory and delivered medications to patients in the field.

## Study population

All bacteriologically confirmed RR-TB patients diagnosed during the study period were included. We also enrolled persons with clinical and/or radiological evidence of possible TB, even in the absence of bacteriologic confirmation, if the person was a known household contact of a patient with documented RR/MDR-TB.

We included patients with QTcF ≤ 450msec (males) or ≤ 470msec (females), obtained within 14 days prior to the start of the study treatment. We excluded patients with known allergies or hypersensitivities to two or more drugs in the proposed treatment regimen.

## Study variables and definitions

The 2013 WHO outcome definitions were used. Cure and treatment completion were categorized as successful outcomes, while treatment failure, death, and loss to follow-up were categorized as unsuccessful outcomes.

Culture conversion was defined as a participant who had a positive culture at baseline and had two consecutive negative cultures taken at least 14 days apart (+/− 7 days) with no intervening positive cultures. Time to culture conversion was defined as the time between treatment initiation and the time-point of the specimen collection of the first negative culture.

Adverse Events were graded on a scale of 1–4, with 1 being mild and 4 being life-threatening. The grading followed the National Institute of Allergy and Infectious Diseases (NIAID) Division of Microbiology and Infectious Diseases (DMID) grading system and a selection of relevant terms from the National Cancer Institute (NCI) Common Terminology Criteria for Adverse Events (CTCAE) [9,10]. Serious adverse events (SAE) were defined as any adverse event that resulted in either death, permanent/significant disability or incapacity, hospitalization or prolongation of hospitalization to manage the adverse event or was categorized as life-threatening. We used Hy's criteria for liver dysfunction, which includes an ALT and AST of more than three times the upper limits of normal and a total bilirubin of more than two times the upper limit of normal and no other reason can be found to account for these elevations. A permanent change in the treatment regimen was defined as the decision by the consilium to stop one or more medications in the treatment regimen for more than 30 days.

Hazardous and harmful drinking was defined and measured using the Alcohol Use Disorders Identification Test (AUDIT) followed by evaluation by a psychologist and a psychiatrist [11]. Hazardous and harmful drinking was defined as AUDIT scores of 8–13 for females and 8–15 for males and >13 for females and >15 for males, respectively. For WHOQoL-BREF and PHQ-9 assessments, the baseline was defined as the first measurement within 30 days before or after treatment initiation. The end of treatment was defined as an assessment 30 days before or after the end of treatment.

### Data collection and analysis

All patient information was collected using standardized study forms and encoded into EndTB Bahmni®. Descriptive analysis was carried out to describe the characteristics of the study population using proportions, means with standard deviation, or median with interquartile range. For the analysis of time to culture conversion, the outcome of interest was culture conversion achieved at six months of treatment (as a categorical variable). Log-binomial regression analysis was used to explore associations between selected demographic and clinical characteristics and unfavorable outcomes. We included variables with a p-value <0.2 in the initial multivariable model, as well as factors that made sense clinically, even if their p-value was slightly above 0.2. We then refined the model using backward elimination and stepwise selection. For the estimation of post-treatment recurrence, we assumed different scenarios about deaths, and we excluded patients with missing follow-up data [12]. First, we included patients who died but did not count them as recurrences, thereby assuming that patients who died did not experience recurrent TB. Second, we included patients who died and we counted them as recurrences, thereby assuming that patients who died experienced recurrent TB. Third, we excluded patients who died from the calculations. In all the scenarios, we treated missing FU as missing data and excluded these patients. WHOQoL-BREF scores were calculated as per the WHO instructions.

We used a paired t-test for the analysis of the WHOQOL-BREF data. Data were analyzed using STATA version 15 (StataCorp, College Station, TX, USA).

### Ethics

Ethics approval was obtained by MSF Ethics Review Board and the Medical Ethics committee of "National Institute of Physiology and Pulmonology of the National Academy of Medical Sciences of Ukraine". We sought written informed consent among persons ≥ 18 years or guardian consent and participant assent for persons ≥ 12 years of age or only guardian consent for < 12 years of age.

## Results

### Socio-demographic and clinical characteristics

From 1 April 2019–31 May 2021, 300 patients were enrolled in the study with a median age of 45 years (IQR 38–52 years). Table 1 shows the key characteristics of these patients. Among these, 233 (78%) were male, 69 (23%) were living with HIV, 6 (2%) had Hepatitis B, 60 (20%) with history or active Hepatitis C (HCV), 13 (4%) had diabetes mellitus, 204 (69%) smoked cigarettes, 14 (5%) persons who injected drugs (PWID), 68 (24%) had hazardous drinking, and 73 (25%) had alcohol dependence. Furthermore, 168 (56%) participants had previously been treated with first or second-line drugs (Table 1).

### End of treatment outcomes

Overall, 78% of patients had successful treatment outcomes. The mortality was 13%, with a median duration to death from treatment initiation of 2.1 months (IQR 0.4 – 4.3 months). The lowest treatment success was observed among PWIDs (64%) (Tables 2–3). There was no statistically significant difference among successful outcomes for those with AUD (p = 0.785), HCV+ (p = 0.430) or PLHIV (p = 0.539) compared to those with no comorbid conditions.

Patients who were previously treated with first-line drugs (75%, p = 0.033) or second-line drugs (66%, p < 0.001) had significantly lower successful treatment outcomes than those who were never treated for TB (86%) earlier.

### 12-month post-treatment outcomes

Among the 234 patients with successful treatment outcomes, 172 (74%) were recurrence-free at 12 months post-treatment. Seventeen patients (7%) died post-treatment with a median time to death of 7.9 months (IQR: 6.0 – 11.7 months) post-treatment. Cause of death was ascertained for all patients, and none were related to complications of TB disease or treatment.

**Table 1. Baseline socio-demographic characteristics, co-morbidities, substance use, TB treatment history, and DRTB drug resistance profile among RR-TB patients in Zhytomyr Oblast, Ukraine, April 2019 – May 2021.**

| Characteristics | Total (n = 300) |
|---|---|
| | n (%) |
| **Socio-demographic characteristics** | |
| Sex (Male) | 233 (78) |
| Age in years (Median [IQR]) | 45 [38-52] |
| HIV+ | 69 (23) |
| HBV+ | 6 (2) |
| HCV+ (history or active) | 60 (20) |
| Diabetes | 13 (4) |
| Smoked cigarettes | 204 (68) |
| Persons who inject drugs (PWID) | 14 (5) |
| Hazardous drinking[1] | 68 (24) |
| Alcohol dependence[2] | 73 (25) |
| **TB treatment history** | **n = 300** |
| No previous TB treatment | 127 (42) |
| Previously treated with first line drug | 93 (31) |
| Previously treated with second line drug | 75 (25) |
| TB treatment history unclear/unknown | 5 (2) |
| **Sub-class drug resistance profile** | **n = 269** |
| Poly-resistance[3] | 3 (1) |
| Mono-resistance | 25 (9) |
| Confirmed MDR | 85 (32) |
| pre-XDR[4] | 156 (58) |

[1]Hazardous drinking: (Female: Audit≥8–13; Male: Audit≥8–15) [2]Harmful drinking: (Female: Audit>13; Male: Audit>15) [3]Poly-resistance = one H(S) resistance and 2 HE(S) resistance. [4]pre-XDR: following the 2020 definition of pre-XDR TB, resistance to R, H and either a fluoroquinolone or aminoglycoside.

**Table 2. End of treatment outcome by comorbidities among RR-TB patients in Zhytomyr Oblast, Ukraine.**

| Treatment outcome | Total[6] (n = 300) | HIV + (n = 69) | Hep C[1] (n = 60) | PWID[2] (n = 14) | Hazardous drinking[3] (n = 68) | Alcohol dependence[4] (n = 73) | No comorbid/risk factor[5] (n = 99) |
|---|---|---|---|---|---|---|---|
| | n (%) | n (%) | n (%) | n (%) | n (%) | n (%) | n (%) |
| Cured | 212 (71) | 50 (72) | 38 (63) | 9 (64) | 41 (60) | 50 (69) | 63 (64) |
| Completed | 22 (7) | 7 (10) | 6 (10) | 0 | 10 (15) | 8 (11) | 15 (15) |
| Died | 40 (13) | 8 (12) | 10 (17) | 4 (29) | 10 (15) | 9 (12) | 11 (11) |
| Treatment failure | 17 (6) | 3 (5) | 5 (8) | 1 (7) | 6 (9) | 2 (3) | 7 (7) |
| LTFU | 9 (3) | 1 (1) | 1 (2) | 0 | 1 (1) | 4 (5) | 3 (3) |
| Treatment success | 234 (78) | 57 (82) | 44 (73) | 9 (64) | 51 (75) | 58 (80) | 78 (79) |

[1]Hepatitis C active or history; [2]PWID = People Who Inject Drugs; [3]Hazardous drinking: Female: Audit≥8–13; Male: Audit≥8–15; [4]Alcohol dependence: Female: Audit>13; Male: Audit>15; [5]No comorbidities/risk factor = HIV negative, Hepatitis B non-reactive, no active or history of Hepatitis C, not PWID, and not hazardous drinker/alcohol dependence. [6]Comorbidites are not mutually exclusive.

**Table 3. End of treatment outcome by history of treatment and resistance profile among RR-TB patients in Zhytomyr Oblast, Ukraine.**

| Treatment outcome | Previous treatment | | | Drug resistance | | |
|---|---|---|---|---|---|---|
| | Never treated/unclear history (n = 132) | Prev. 1st line (n = 93) | Prev. 2nd line (n = 75) | Poly and mono resistance (n = 28) | Confirmed MDR (n = 85) | Pre-XDR (n = 156) |
| | n(%) | n (%) | n (%) | n (%) | n (%) | n (%) |
| Cured | 101 (76) | 66 (71) | 45 (60) | 22 (79) | 61 (72) | 108 (69) |
| Completed | 13 (10) | 4 (4) | 5 (6) | 1 (3) | 4 (5) | 9 (6) |
| Died | 11 (8) | 15 (16) | 14 (19) | 2 (7) | 13 (15) | 24 (16) |
| Treatment failure | 2 (<1) | 6 (7) | 9 (12) | 2 (7) | 4 (5) | 10 (6) |
| LTFU | 5 (<1) | 2 (2) | 2 (3) | 1 (3) | 3 (3) | 5 (3) |
| Success rate | 114 (86) | 70 (75) | 50 (66) | 23 (82) | 65 (76) | 117 (75) |

One patient (<1%) experienced recurrent TB during the post-treatment follow-up (Table 4). Two additional patients, who were culture negative 12 months post treatment, were deemed to have recurrent TB by clinical diagnosis.

The alternative estimates of post-treatment recurrence are presented in Table 5. The estimates for recurrence-free post-treatment survival at 12 months post-treatment range from 91% to 99%.

## Factors associated with end-of-treatment outcomes

Multivariate regression analysis looking at the association between selected clinical and sociodemographic characteristics and unfavorable outcomes (death and LTFU), has shown that patients with a history of previous treatment with 1st or 2nd line drugs had a 2.38 times higher risk of unfavorable treatment outcomes compared to those with no history of previous treatment (95% CI 1.30 – 4.39). (Table 6).

## Culture conversion

There were 249 patients with a positive culture result at baseline. Out of these patients, 45 died or were LTFU before six months and were excluded from the 6-month culture conversion analysis. Of the 204 patients, only one

**Table 4. Overall 12-month post treatment outcomes among RR-TB patients in Zhytomyr Oblast, Ukraine.**

| Treatment outcome | Total (n = 234) | Cured (n = 212) | Completed (n = 22) |
|---|---|---|---|
| | n (%) | n (%) | n (%) |
| Recurrence free and alive at 12 months post-treatment | 174 (74) | 161 (75) | 13 (59) |
| Died post treatment | 17 (7) | 15 (7) | 2 (9) |
| Relapse or recurrence | 1 (<1) | 1 (<1) | 0 (0) |
| LTFU post treatment | 42 (18) | 35 (17) | 7 (32) |

**Table 5. Alternate estimates of 12-month post treatment recurrence RR-TB patients in Zhytomyr Oblast, Ukraine.**

| Treatment outcome | Scenario 1 (n = 192) | Scenario 2 (n = 192) | Scenario 3 (n = 175) |
|---|---|---|---|
| | n (%) | n (%) | n (%) |
| Recurrence free and alive at 12 months post-treatment | 174 (91) | 174 (91) | 174 (99) |
| Died post treatment | 17 (9) | 18 (9) | excluded |
| Relapse or recurrence | 1 (<1) | | 1 (1) |
| LTFU post treatment | excluded | excluded | Excluded |

Scenario 1: included patients who died but did not count them as recurrences; Scenario 2: included patients who died as recurrent TB; Scenario 3: excluded patients who died from the calculations.

Global Public Health

**Table 6. Multivariate analysis risk factors for unfavorable outcome among RR/MDR-TB in Zhytomyr Oblast, Ukraine.**

| Characteristic | Univariate | | | Multivariate | | |
|---|---|---|---|---|---|---|
| | Risk Ratio | 95% Conf. Interval | p-Value | Adjusted Risk Ratio | 95% Conf. Interval | p-Value |
| Sex | | | | | | |
| Male | 1 | | | 1 | | |
| Female | 0.64 | 0.31 – 1.30 | **0.213** | 0.71 | 0.34 – 1.48 | 0.368 |
| Age (years) | | | | | | |
| 0–44y | 1 | | | 1 | | |
| 45y+ | 1.45 | 0.84 – 2.52 | **0.184** | 1.48 | 0.84 – 2.61 | 0.172 |
| HIV status | | | | | | |
| Negative | 1 | | | | | |
| Positive | 0.69 | 0.34 – 1.37 | 0.287 | | | |
| Hepatitis C status | | | | | | |
| No history | 1 | | | | | |
| History/active | 1.38 | 0.72 – 2.65 | 0.331 | | | |
| Hepatitis B | | | | | | |
| Non-reactive | 1 | | | | | |
| Reactive | 1.80 | 0.32 – 10.03 | 0.504 | | | |
| Diabetes | | | | | | |
| No | 1 | | | | | |
| Yes | 1.07 | 0.28 – 3.99 | 0.924 | | | |
| Smoked cigarettes | | | | | | |
| No | 1 | | | | | |
| Yes | 1.10 | 0.60 – 2.01 | 0.752 | | | |
| Injectable drug use | | | | | | |
| No | 1 | | | 1 | | |
| Yes | 2.05 | 0.66 – 6.34 | **0.213** | 1.61 | 0.50 – 5.17 | 0.424 |
| Alcohol consumption | | | | | | |
| Low consumption | 1 | | | | | |
| Hazardous and addiction | 1.20 | 0.68 – 2.12 | 0.519 | | | |
| History of previous treatment | | | | | | |
| No treatment history | 1 | | | 1 | | |
| Previously treated with 1st or 2nd line drugs | 2.53 | 1.39 – 4.61 | **0.002** | 2.38 | 1.29 – 4.38 | **0.005** |
| Drug resistance profile | | | | | | |
| Mono and Poly-resistance | 1 | | | | | |
| MDR TB | 1.42 | 0.48 – 4.21 | 0.532 | | | |
| Pre-XDR TB | 1.53 | 0.54 – 4.31 | 0.417 | | | |

patient did not have culture conversion at six months. Overall, 50% of patients with a positive culture at baseline had culture conversion within 60 days of treatment initiation. The median time to culture conversion was 58 days (IQR 30–75).

## Safety of the regimen

A total of 122 AEs were recorded. Most adverse events 37 (30%) were related to increased liver enzymes. Of our patients, 86/300 (29%) experienced at least one AE grade 3 or 4. The median number of grade 3–4 AEs per patient was 2 (IQR 1–2).

**Table 7. Proportion of patients with adverse events of interest among RR-TB patients in Zhytomyr Oblast, Ukraine.**

| Type of adverse events (at least once) | Total (n = 300) |
|---|---|
| | n (%) |
| Serious adverse events[1] | 63 (21) |
| Grade 3 or 4 adverse events | 86 (27) |
| QTc prolongation of any grade | 61 (20) |
| Optic neuritis of any grade | 5 (2) |
| Meeting Hy's criteria for liver dysfunction[2] | 46 (15) |
| Permanent change in treatment regimen[3] | 79 (26) |

[1]Any adverse events or reaction that results in death, permanent/significant disability or incapacity, hospitalization or prolongation of hospitalization to manage the adverse event, or is life-threatening.

[2]Defined as an ALT or AST more than 3 times the upper limits of normal, or a total bilirubin more than 2 times the upper limit of normal and no other reason can be found to account for these elevations.

[3]Defined as the decision by a study physician to stop one or more medications in the treatment regimen for more than 30 days.

**Table 8. Comparison of BREF WHO QOL at baseline and treatment completion (n = 180) among RR-TB patients in Zhytomyr Oblast, Ukraine.**

| Domain | Baseline | End of treatment | t-test p-value |
|---|---|---|---|
| | Mean [Std. Dev] | Mean [Std. Dev] | (for no difference) |
| Physical health | 55.39 (14.97) | 55.24 (12.58) | 0.92 |
| Psychological | 64.13 (16.18) | 64.4 (15.92) | 0.86 |
| Social relationship | 59.49 (24.31) | 63.47 (25.20) | 0.13 |
| Environment | 62.41 (16.29) | 65.78 (16.69) | **0.05** |
| Overall | 52.64 (21.63) | 57.15 (21.43) | **0.05** |

A total of 67 serious AEs were reported. Of our patients, 63/300 (21%) experienced at least one serious AEs. The full list of adverse events of interest is in Table 7. The type of drug that led to permanent change, the reason for the change and the outcomes of patients who underwent changes are presented in S1 Table, S2 Table and S3 Table, respectively).

### BREF WHO QOL and PHQ9

To conduct the BREF WHO QOL and PHQ 9 analysis, we excluded 30 patients who had outcomes within the first three months after starting treatment and four patients under 17.

Overall, while the WHOQoL scores were low, there was a significant increase in the overall score at the end of treatment compared to baseline (p < 0.05) (Table 8). However, we did not observe any significant difference in the physical health and psychological domains at the end of treatment.

We observed a decline in the PHQ-9 scores from the baseline to the end of treatment. The average PHQ-9 score decreased significantly (p = 0.04) from 6.67 (standard deviation (SD) 4.75) at baseline to 5.34 (SD: 5.18) at the end of treatment.

### Discussion

Our single-arm, operational research study of an RR/MDR-TB package of care, including a shorter, all-oral regimen recommended by WHO in June 2024 [13]. achieved high treatment success rates, even under stressful program conditions.

These successful outcomes were sustained over a 12-month follow-up period after treatment completion. However, there was notable LTFU in the post-treatment period, as well as death and recurrent TB. That these results were achieved during the COVID-19 pandemic and a period of active war in Ukraine, suggests a high functional utility of our approach over a range of program settings.

The landscape for treating RR/MDR-TB has changed dramatically in the past five years. There are now multiple randomized controlled trials showing the utility of all-oral regimens lasting six to nine months. Some of these studies used regimens with a high pill burden [14]. Others utilized drugs that are not commonly available in programmatic settings, such as pretomanid or delamanid [15–17]. In addition, treatment regimens used in randomized controlled trials tend to perform better than those same regimens used under program conditions and in populations that would normally be excluded from formal trials. Our results are similar to those seen in other cohort studies assessing all-oral shorter regimens for RR/MDR-TB[1] [18–22].

In addition to the all-oral shorter regimens used in our study, we offered a comprehensive package of patient support services beyond those routinely provided by the national TB program, including counseling, intensive nursing care, management of AUD and other comorbid conditions, and nutritional and social support. Our goal in offering these supporting services was to provide a person-centered experience for the patients being treated [23]. Our study found successful treatment outcomes among patients with AUD, people living with HIV and other comorbid conditions, along with significant improvements in quality of life and depression measures, pointing to the success of our inclusive approach. While much attention has been given to newer drugs and shorter regimens, offering psychosocial and socioeconomic support during RR/MDR-TB is likely to be just as important, especially under program conditions, where high rates of LTFU during treatment are still being reported despite the use of shorter regimens [24]. Given the high rates of substance use and other co-morbidities seen in our cohort, such support may have been even more important [25]. The impact of such support was likely even higher in the times of COVID-19 and the ongoing war with Russia. Our study methodology did not allow for separating the impact of the support package from the drug regimen on treatment success rates, and this is a limitation. We determined, however, that there was value in combining the two approaches from a humanitarian and programmatic point of view.

Although our operational research project showed high success rates overall, there were some worrisome findings. First, our study continued to show high rates of toxicity, with more than 20% of our patients experiencing a serious adverse event. Although these rates are similar to those reported in other studies, they emphasize the need for regimens that are both safer and better tolerated by people undergoing treatment for RR/MDR-TB. Second, there was a high rate of LTFU in the post-treatment period, where 18% of people with successful treatment outcomes at the end of treatment could not be assessed for long-term outcomes. This may have been due to disruptions in the country due to COVID-19 and the war, but it is possible that these patients may also have developed recurrent TB or died. In fact, when the post-treatment period is considered, only 172 of 300 patients (57.3%) could be confirmed as having treatment success. Most TB programs do not continue to follow patients after they have successfully completed treatment, even though the WHO recommends such follow-up. Our results suggest that investment in such follow-up is needed to determine the long-term impact of RR/MDR-TB treatment.

## Conclusion

In conclusion, we report a high rate of successful treatment outcomes along with an improvement in the quality of life of patients amidst challenging conditions of the COVID-19 pandemic and war in Ukraine through the implementation of a person-centered model of care with all-oral short-course regimen. With the encouraging results from multiple randomized controlled trials with short course regimens it is time to also focus on developing person-centered models of care to deliver these regimens in a manner that, along with achieving successful treatment outcomes, leads to an improvement in the quality of life of people with RR-TB.

## Supporting information

**S1 Table. Type of drugs with permanent change (interruption >30 days) among RR-TB patients in Zhytomyr Oblast, Ukraine, April 2019 – March 2022.**
(DOCX)

**S2 Table. Reason of permanent drug changes (interruption >30 days) among RR-TB patients in Zhytomyr Oblast, Ukraine, April 2019 – March 2022.**
(DOCX)

**S3 Table. Outcome of patients with at least one permanent change on their regimen (interruption >30 days) among RR-TB patients in Zhytomyr Oblast, Ukraine, April 2019 – March 2022.**
(DOCX)

**S1 Checklist. Inclusivity in global research.**
(DOCX)

**S1 File. STROBE Checklist.**
(DOCX)

## Author contributions

**Conceptualization:** Olena Trush, Nataliia Lytvynenko, Khachatur Malakyan, Yves Wally, Jennifer Furin, Dmytro Donchuk, Petros Isaakidis.

**Data curation:** Vini Fardhdiani, Svitlana Pylypchuk, Praharshinie Rupasinghe, Chinmay Laxmeshwar, Petros Isaakidis.

**Formal analysis:** Vini Fardhdiani, Chinmay Laxmeshwar, Petros Isaakidis.

**Investigation:** Olena Trush, Nataliia Lytvynenko, Yana Terleeva, Khachatur Malakyan, Praharshinie Rupasinghe, Marve Duka, Vitaly Stephanovich Didyk, Olga Valentinovna Siomak, Oleksandr Blyzniuk.

**Methodology:** Nataliia Lytvynenko, Jennifer Furin, Chinmay Laxmeshwar, Petros Isaakidis.

**Project administration:** Olena Trush, Svitlana Pylypchuk, Yana Terleeva, Khachatur Malakyan, Yves Wally, Marve Duka, Vitaly Stephanovich Didyk, Olga Valentinovna Siomak, Oleksandr Blyzniuk, Petros Isaakidis.

**Resources:** Olena Trush, Nataliia Lytvynenko, Svitlana Pylypchuk, Yana Terleeva, Khachatur Malakyan, Praharshinie Rupasinghe, Yves Wally, Marve Duka, Vitaly Stephanovich Didyk, Olga Valentinovna Siomak, Oleksandr Blyzniuk, Jennifer Furin, Dmytro Donchuk, Petros Isaakidis.

**Software:** Vini Fardhdiani.

**Supervision:** Yana Terleeva, Yves Wally, Marve Duka, Vitaly Stephanovich Didyk, Dmytro Donchuk, Petros Isaakidis.

**Validation:** Nataliia Lytvynenko, Svitlana Pylypchuk, Praharshinie Rupasinghe, Vitaly Stephanovich Didyk, Chinmay Laxmeshwar, Petros Isaakidis.

**Visualization:** Petros Isaakidis.

**Writing – original draft:** Vini Fardhdiani, Petros Isaakidis.

**Writing – review & editing:** Vini Fardhdiani, Olena Trush, Nataliia Lytvynenko, Svitlana Pylypchuk, Yana Terleeva, Khachatur Malakyan, Praharshinie Rupasinghe, Yves Wally, Marve Duka, Vitaly Stephanovich Didyk, Olga Valentinovna Siomak, Oleksandr Blyzniuk, Jennifer Furin, Dmytro Donchuk, Chinmay Laxmeshwar, Petros Isaakidis.

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
