## [Decision Letter · Decision Letter 0]

11 Feb 2025

PGPH-D-24-02638

End of treatment and 12-month post-treatment outcomes in patients treated with all-oral regimens for rifampicin-resistant tuberculosis in Ukraine

Dear Dr. Isaakidis,

Thank you for submitting your manuscript to PLOS Global Public Health. After careful consideration, we feel that it has merit but does not fully meet PLOS Global Public Health’s publication criteria as it currently stands. Therefore, we invite you to submit a revised version of the manuscript that addresses the points raised during the review process.

We look forward to receiving your revised manuscript.

Kind regards,

N. Sarita Shah

Academic Editor

Journal Requirements:

Reviewers' comments:

Reviewer's Responses to Questions

**Comments to the Author**

1. Does this manuscript meet PLOS Global Public Health’s publication criteria ? Is the manuscript technically sound, and do the data support the conclusions? The manuscript must describe methodologically and ethically rigorous research with conclusions that are appropriately drawn based on the data presented.

Reviewer #1: Yes

Reviewer #2: Yes

2. Has the statistical analysis been performed appropriately and rigorously?

Reviewer #1: Yes

Reviewer #2: Yes

3. Have the authors made all data underlying the findings in their manuscript fully available (please refer to the Data Availability Statement at the start of the manuscript PDF file)?

Reviewer #1: No

Reviewer #2: Yes

4. Is the manuscript presented in an intelligible fashion and written in standard English?

Reviewer #1: No

Reviewer #2: Yes

5. Review Comments to the Author

Reviewer #1: Overall, this is a well conducted observational study assessing person level outcomes for the treatment of MDR/RR-TB using oral, short treatment regimens. Given the difficult context in which the study was conducted, these data are important to demonstrate how such treatment can be provided and good outcomes achieved. The study is therefore relevant to a wide reader audience.

Major comment:

The manuscript would be greatly improved by following STROBE guidance for presentation of observational studies; see https://www.equator-network.org/reporting-guidelines/strobe/ This would include clearer presentation of the baseline characteristics of the study cohort and a flow diagram describing eligibility, inclusion and exclusion for the study.

Other comments:

• Lines 108-109: there is mention of treatment education, but a much more detailed description is given later – perhaps to exclude here?

• Lines 110-112: the section describing the provision of ambulatory care as different from national policy could be clarified.

• Line 147: how was adherence assessed?

• Define acronyms at first use, eg. PWID

• Box 1: How was the community engaged to provide support to patients and families?

• Under results, there appear to be 2 sections titled “end of treatment outcomes”.

• Please revise language to refer to ‘treatment failure” rather than “failed”, which implies that the individuals themselves have failed in some way.

• Were any post-treatment outcomes available for people who had treatment outcomes of “treatment failure” or LTFU? I see this is described in the discussion, but could be presented in the results.

• For table 5, please add a description of each scenario to the table legend.

• For the time to culture conversion analysis, was it not possible to censor for death and LTFU? Rather than exclude these patients completely? Additionally, you report time to culture conversion by alcohol use, but no other factors; were any other relevant factors significant? Eg. extent of drug resistance or previous TB treatment?

• Perhaps to clarify why pregnancy during treatment might be considered an adverse event?

• The methods mention deep sequencing, but no data pertaining to this is presented. What was the purpose of deep sequencing of TB isolates?

• To clarify that the majority of patients would have completed treatment by the time of the Russian invasion of Ukraine in Feb 2022 – does this mean that the modified treatment provisions were perhaps only relevant for the post-treatment period for most patients?

Reviewer #2: General Comments:

The treatment of rifampin-resistant TB (RR-TB) has changed dramatically over the past 10 years—most notably with the change from 18-24 month injectable-based regimens to 6-9 month all-oral regimens incorporating new and repurposed medications. Many prior cohort studies have reported RR-TB treatment outcomes from a variety of programmatic settings around the world. Context is often key, however, when interpreting and comparing results from different countries as some settings lack access to newer diagnostics or medications, while others may partner with NGOs to provide additional patient-centered support.

In this manuscript, authors from Medecins Sans Frontieres report their experience supporting an RR-TB program in Ukraine over a time period spanning both the emergence of the COVID pandemic and the start of the war with Russia.

Clearly, the real-life challenges to providing (and completing) RR-TB care in this setting are vast, yet the authors demonstrate that they were able to achieve startlingly favorable end-of-treatment outcomes. The paper is very well-written and clear (though there are a number of typographical and formatting errors). Importantly, the authors emphasize not only the mechanics of providing treatment, but also of supporting patients socially to ensure treatment completion. Participants were administered quality-of-life questionnaires. Of particular value is that the authors also captured post-treatment outcomes 12-months after treatment completion. They note high rates of LTFU in this post-treatment period and raise the possibility that relapse/death could be much higher in the long-term, even if end-of-treatment outcomes were favorable. The data are a valuable contribution to the literature and should serve as a model to others trying to set up or maintain RR-TB services in conflict-areas.

Specific Comments:

ABSTRACT

1) Line 47, Results: Suggest delete the word “and.”

INTRODUCTION

1) Why are dates of enrollment are highlighted throughout the manuscript?

2) Line 98: “prevalence…” Suspect that the authors mean “incidence” here.

3) Line 100: Would consider using a word other than “Dispensary.” Although this is what they are called in Ukraine, the term “dispensary” in the rest of the world more often refers to a pharmacy. Could say “TB referral hospital” or “TB specialist hospital/center” instead.

4) Line 100: “The dispensary also coordinated the ambulatory care…” How so? Do patients go there for outpatient care or does the dispensary arrange for patients to be seen at some decentralized clinic instead?

5) Line 114: “…outpatient facilities…” See comment above. How close are these facilities? Are they spread out over a wide geographic area? Close to patients’ homes?

METHODS

1) No methods are provided for the logistic regression presented in the Results. How were covariates selected for the multivariable model?

RESULTS

1) Line 244: “End…” is not bolded while the rest of the heading is.

2) Line 247: “(Table & 3)”: Typo

3) Lines 250-1: The placement of the percentages and p-values in the sentences is confusing.

4) Lines 255-6: “Of our patients, 233 (78%) were male, had a median age of 45 years…” Grammatical error. Would rephrase.

5) Line 258: Would avoid the non-person-centered term “smoker” (suggest “smoked cigarettes”).

6) Line 260: “Error! (Reference source not found)”—Likely typographical error.

7) Line 261-269: This paragraph appears to have been completely duplicated ( lines 244-252)

8) Line numbers start again from 1 halfway through the RESULTS.

9) Second line 4: Suggest “…with a median time to death of 7.9 months…”

10) Second lines 34-6: “…45 died or were LTFU before six months and were excluded from the 6-month culture-conversion analysis.” There are many different philosophies about the best approach in the longitudinal analysis of culture conversion. Excluding those who die or who are LTFU clearly introduces a survival bias to even be eligible for the analysis. One could argue that Death prior to culture conversion is, itself, strongly associated with NOT converting. That said, not everyone who is LTFU necessary fails to convert (just as not everyone who dies does so from TB) so including them in the analysis may negatively skew the results. Regardless, however, the authors should simply provide a clear justification of their approach and explain why their approach will minimize certain biases while acknowledging that it may introduce others.

11) Second line 49: Would suggest providing the number of participants as well as the total number of AEs.

12) Second Line 52: “experience” should be “experienced.”

13) Second Line 53: “The full list of…” Sentence appears to have been truncated erroneously.

14) Second Line 54: “Table.” (Is there a table missing? Not numbered. Typo?)

DISCUSSION

1) Second line 117: Suspect that the authors mean “serious” rather than “severe.”

TABLES/FIGURES/BOXES

Box 1:

1) The verb tense varies in the “expected outcomes” column (e.g., “Patients realize….” vs. “Patients learned….” vs. “Family members understand….” Would encourage consistent use of one tense (probably present or future if this an “expected” rather than observed outcome).

2) Define abbreviations in the Box (AUD, SUD, VOT, SAT, etc).

3) In the last section (“Community Support”), part of “Involving communities” is bolded in error.

Table 1

4) See earlier comment about the term “smoker.”

Table 2

5) The number 6 in the footnotes should be a superscript.

Table 6

6) How were characteristics selected for inclusion in the multivariable analysis? Three of the four included covariates had a p-value >0.1 even in bivariate analysis.

Table 7

7) Second line 60: Footnote 2: Suggest deleting the word “which.”

8) Second line 62: Suggest deleting the first clause and beginning with “Defined as the decision…”

9) In this table it says that 79 participants had a permanent change in regimen due to AEs, but in supplemental table S2 this number is only 13.

Table 8

10) Footnotes are not provided for 1-5?

Supplementary Table S1

11) Although mention is made in the Methods about “Permanent changes.,” the supplementary tables are not referenced anywhere in the main text.

12) Hard to interpret this table without knowing how many patients got each drug. It looks as though levofloxacin is very poorly tolerated. It might be more helpful if Tables S1 and S2 were combined so that one could see WHY each drug was stopped (I suspect that levofloxacin was stopped for resistance, while clofazimine was stopped for side effects).

13) What is the denominator for this table? Participants? Why only 95?

6. PLOS authors have the option to publish the peer review history of their article (what does this mean? ). If published, this will include your full peer review and any attached files.

**Do you want your identity to be public for this peer review?** For information about this choice, including consent withdrawal, please see our Privacy Policy .

Reviewer #1: No

Reviewer #2: No

---

## [Editor Report · Decision Letter 1]

24 Apr 2025

End of treatment and 12-month post-treatment outcomes in patients treated with all-oral regimens for rifampicin-resistant tuberculosis in Ukraine: a prospective cohort study

PGPH-D-24-02638R1

Dear Dr. Isaakidis,

We are pleased to inform you that your manuscript 'End of treatment and 12-month post-treatment outcomes in patients treated with all-oral regimens for rifampicin-resistant tuberculosis in Ukraine: a prospective cohort study' has been provisionally accepted for publication in PLOS Global Public Health.

Best regards,

N. Sarita Shah

Academic Editor
